# DISCOVERING GENERAL-PURPOSE ACTIVE LEARNING STRATEGIES

## ABSTRACT

We propose a general-purpose approach to discovering active learning (AL) strategies from data. These strategies are transferable from one domain to another and can be used in conjunction with many machine learning models. To this end, we formalize the annotation process as a Markov decision process, design universal state and action spaces and introduce a new reward function that precisely model the AL objective of minimizing the annotation cost We seek to find an optimal (non-myopic) AL strategy using reinforcement learning. We evaluate the learned strategies on multiple unrelated domains and show that they consistently outperform state-of-the-art baselines.

## 1 INTRODUCTION

Modern supervised machine learning (ML) methods require large annotated datasets for training purposes and the cost of producing them can easily become prohibitive. Active learning (AL) mitigates the problem by selecting intelligently and adaptively a subset of the data to be annotated. To do so, AL typically relies on informativeness measures that identify unlabelled datapoints whose labels are most likely to help to improve the performance of the trained model. As a result, good performance is achieved using far fewer annotations than by randomly labelling data.

Most AL selection strategies are hand-designed either on the basis of researcher's expertise and intuition or by approximating theoretical criteria (Settles, 2012). They are often tailored for specific applications and empirical studies show that there is no single strategy that consistently outperforms others in all datasets (Baram et al., 2004; Ebert et al., 2012). Furthermore, they only represent a small subset of all possible strategies.

To overcome these limitations, it has recently been proposed to design the strategies themselves in a data-driven fashion by learning them from prior experience with AL (Konyushkova et al., 2017; Bachman et al., 2017). This *meta* approach makes it possible to go beyond human intuition and potentially to discover completely new strategies by accounting for the state of the trained ML model when selecting the data to annotate. However, many of these methods are still limited to either learning from closely related domains (Bachman et al., 2017; Fang et al., 2017; Liu et al., 2018), or using a greedy selection that may be suboptimal (Konyushkova et al., 2017; Liu et al., 2018), or relying on properties of specific classifiers (Konyushkova et al., 2017; Bachman et al., 2017; Contardo et al., 2017; Ravi & Larochelle, 2018). In short, even though data-driven AL methods have flourished recently, there is still no *general-purpose* non-myopic methods that depend neither on the kind data nor on the specific ML model used in training. In this paper, we introduce such a generic data-driven AL method that is applicable to heterogeneous datasets and to most ML models because it does not require hand-crafting model- or dataset-specific features.

More specifically, we reformulate AL as a Markov Decision Process (MDP) and use reinforcement learning (RL) to find AL strategy as an optimal MDP policy. To achieve the desired generality, we incorporate two important contributions into our approach. First, we take the AL objective to be minimizing the number of annotations required to achieve a given prediction quality, which is a departure from standard AL approaches that maximize the performance given an annotation budget. In this way, we optimise what the practitioners truly want, that is, the annotation cost, independently of the specific ML model and performance measure being used. To this end, we design the reward function of MDP to reflect our AL objective. Second, we propose a procedure that can lean an AL strategy from data coming from multiple unrelated domains for which annotations are already

available. The strategy then applies to domains for which this is not the case. To this end, we defined generic MDP state and action representations that can be computed for arbitrary datasets and without regard to the specific ML model.

In our experiments we demonstrate the effectiveness of our approach for the purpose of binary classification by applying the learned strategies to previously unseen datasets from different domains. We show that they enable us to reach pre-defined quality thresholds with fewer annotations than several baselines, including recent meta-AL algorithms (Hsu & Lin, 2015; Konyushkova et al., 2017). We also analyse the properties of our strategies to understand their behaviour and how it differs from those of more traditional ones.

## 2 RELATED WORK

Manually-designed AL methods differ in their underlying assumptions, computational costs, theoretical guarantees, and generalization behaviours. However, they all rely on a human designer having decided how the data points should be selected. Representative approaches to doing this are uncertainty sampling (Lewis & Gale, 1994), which works remarkably well in many cases (Luo et al., 2013; Sun et al., 2015), query-by-committee, which does not require probability estimates (Gilad-Bachrach et al., 2005; Beluch et al., 2018), and expected model change (Freytag et al., 2014; Käding et al., 2015). However, the performance of any one of these strategies on a never seen before dataset is unpredictable (Baram et al., 2004; Ebert et al., 2012), which makes it difficult to choose one over the other. In this section, we review recent methods to addressing this difficulty.

**Combining AL strategies** If a single manually designed method does not consistently outperform all others, it makes sense to adaptively select the best strategy or to combine them. The algorithms that do it can rely on heuristics (Osugi et al., 2005), on bandit algorithms (Baram et al., 2004; Hsu & Lin, 2015; Chu & Lin, 2016), or on RL to find an MDP policy (Ebert et al., 2012; Long & Hua, 2015). Still, this approach remains limited to combining existing strategies instead of learning new ones. Furthermore, strategy learning happens during AL and its success depends critically on the ability to estimate the classification performance from scarce annotated data.

**Data-driven AL** Recently, the researchers have therefore turned to so-called *data-driven AL* approaches that learn AL strategies from annotated data (Konyushkova et al., 2017; Bachman et al., 2017; Contardo et al., 2017; Ravi & Larochelle, 2018; Liu et al., 2018; Fang et al., 2017; Pang et al., 2018). They learn what kind of datapoints are the most beneficial for training the model given the current state of trained ML model. Then, past experience helps to eventually derive a more effective selection strategy. This has been demonstrated to be effective, but it suffers from a number of limitations. First, this approach is often tailored for learning only from related datasets and domains suitable for transfer or one-shot learning (Liu et al., 2018; Bachman et al., 2017; Fang et al., 2017; Contardo et al., 2017; Ravi & Larochelle, 2018). Second, many of them rely on specific properties of the ML models, be they standard classifiers (Konyushkova et al., 2017) or few-shot learning models (Bachman et al., 2017; Contardo et al., 2017; Ravi & Larochelle, 2018), which restricts their generality. Finally, in some approached the resulting strategy is greedy—for example when supervised (Konyushkova et al., 2017) or imitation learning (Liu et al., 2018) is used—that might lead to suboptimal data selection.

MDP formulation in data-driven AL is used both for *pool-based* AL, where datapoints are selected from a large pool of unlabelled data, and for *stream-based* AL, where datapoints come from a stream and AL decides to annotate a datapoint or not as it appears. In stream-based AL, actions—to annotate or not to—are discrete and Q-learning (Watkins & Dayan, 1992) is the RL method of choice to look for an op (Woodward & Finn, 2016; Fang et al., 2017). By contrast, in pool-based AL, the action selection concerns all potential datapoints that can be annotated and it is natural to characterise them by continuous vectors that makes it not suitable for Q-learning. So, policy gradient (Williams, 1992; Sutton et al., 2000) methods are usually used (Bachman et al., 2017; Contardo et al., 2017; Ravi & Larochelle, 2018; Pang et al., 2018). In this paper we focus on pool-based AL but we would like to reap the benefits of Q-learning that is, lower variance and better data-complexity thanks to bootstrapping. To this end, we take advantage of the fact that although actions in pool-based AL are continuous, their number is finite. Thus, we can adapt Q-learning for our purposes.

Most data-driven AL methods stipulate a specific objective function that is being maximised. However, the methods are not always evaluated in a way that is consistent with the objective that is

optimized. Sometimes, the metric used for evaluation differs from the objective (Bachman et al., 2017; Konyushkova et al., 2017; Pang et al., 2018). Sometimes, the learning objective may include additional factors like discounting (Woodward & Finn, 2016; Fang et al., 2017; Pang et al., 2018) or may combine several objectives (Woodward & Finn, 2016; Contardo et al., 2017). By contrast, our approach uses our evaluation criterion—minimization of the time spent annotating for a given performance level—directly in the strategy learning process.

Among data-driven AL, the approach of Pang et al. (2018) achieves generality by using multiple training datasets to learn strategies, as we do. However, this approach is more complex than ours, relies on policy-gradient RL, and uses a standard AL objective. By contrast, our approach does not require a complex state and action embedding, needs fewer RL episodes for training thanks to using Q-learning, and explicitly maximizes what practitioners care about, that is, reduced annotation cost.

## 3 METHOD

We formulate the AL process as a Markov decision process (MDP) and use reinforcement learning (RL) to find an optimal strategy. In this section, we first outline our design philosophy. We then formalize AL in MDP terms and finally describe our approach to finding an optimal MDP policy. For simplicity, we present our approach in the context of binary classification. However, an almost identical AL problem formulation can be used for other ML tasks and a separate selection policy can be trained for each one.

### 3.1 APPROACH

Our goal is to advance data-driven AL towards general-purpose strategy learning. Desirable strategies should have two key properties. They should be **transferable** across unrelated datasets and have sufficient **flexibility** to be applied in conjunction with different ML models. Our design decisions are therefore geared towards learning such strategies. The iterative structure of AL is naturally suited for an MDP formulation: For every *state* of an AL problem, an *agent* takes an *action* that defines the datapoint to annotate and it receives a *reward* that depends on the quality of the model that is re-trained using the new label. An AL strategy then becomes an MDP *policy* that maps a state into an action.

To achieve seamless transferability and flexibility, our task is therefore to design the states, actions, and rewards to be generic. To this end, we represent states and actions as vectors that are independent from specific dataset feature representations and can be computed for a wide variety of ML models. For example, the probability that the classifier assigns to a datapoint suits this purpose because most classifiers estimate this value. By contrast, the number of support vectors in a support vector machine (SVM) or the number of layers of a neural network (NN) are not suitable because they are model-specific. Raw feature representations of data are similarly inappropriate because they are domain specific.

A classical AL objective is to maximize the prediction quality—often expressed in terms of accuracy, AUC, F-score, or negative squared error—for a given annotation budget. For flexibility's sake, we prefer an objective that is not directly linked to a specific performance measure. We therefore consider the *dual* objective of minimizing the number of annotations required for a given *target quality* value. When learning a strategy by optimizing this objective, the AL agent only needs to know if the performance is above or below this target quality, as opposed to its exact value. Therefore, the procedure is less tied to a specific performance measure or setting. Our MDP reward function expresses this objective by penalizing the agent until the target quality is achieved. This motivates the agent to minimize its "suffering" by driving the amount of requested annotations down.

Having formulated the AL problem as a MDP, we can learn a strategy using RL. We simulate the annotation process on data from a collection of unrelated labelled datasets, that ensures the transferability to new unlabelled datasets. Our approach to finding the optimal policy is based on the deep Q-network (DQN) method of Mnih et al. (2013). To apply DQN with pool-based AL, we modify it in two ways. First, we make it work with MDP where actions are represented by vectors corresponding to individual datapoints instead of being discrete. Second, we deal with the set of actions $\mathcal{A}_t$ that change between iterations $t$ as it makes sense to annotate a datapoint only once.

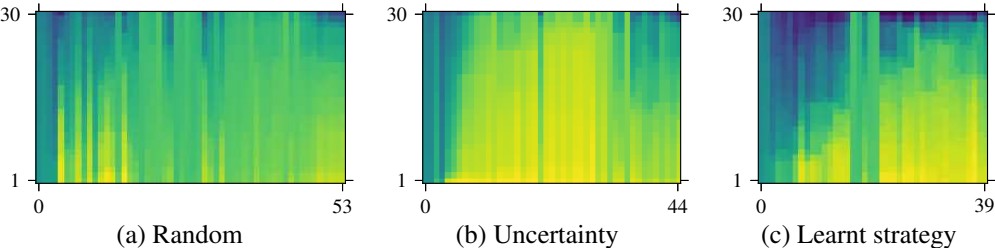

(a) Random  (b) Uncertainty  (c) Learnt strategy

**Figure 1:** The evolution of the learning state vector $s_t$ during an annotation episode starting from the same state for (a) random sampling, (b) uncertainty sampling, and (c) our learnt strategy. Every column represents $s_t$ at iteration $t$, with $|\mathcal{V}| = 30$. Yellow corresponds to values of $\hat{y}_t$ that predict class 1 and blue – class 0.

### 3.2 FORMULATING AL AS AN MDP

Let us consider an AL problem where we annotate a dataset $\mathcal{D}$. A test dataset $\mathcal{D}'$ is used to evaluate the AL procedure. Then, we iteratively select a datapoint $\boldsymbol{x}^{(t)} \in \mathcal{D}$ to be annotated. Let $f_t$ be a classifier trained on a subset $\mathcal{L}_t$ that is annotated after iteration $t$. This classifier assigns a numerical score $\hat{y}_t(\boldsymbol{x}_i) \in \mathbb{R}$ to a datapoint and then maps it to a label $y_i \in \{0, 1\}$, $f_t : \hat{y}_t(\boldsymbol{x}_i) \mapsto \hat{y}_i$. For example, if the score is the predicted probability $\hat{y}_t(\boldsymbol{x}_i) = p(y_i = 0 | \mathcal{L}_t, \boldsymbol{x}_i)$, the mapping function simply thresholds it at $0.5$. If we wanted to perform a regression instead, $\hat{y}_t(\boldsymbol{x}_i)$ could be a predicted label and the mapping function would be the identity. In AL evaluation we measure the quality of classifier $f_t$ by computing its empirical performance $\ell_t$ on $\mathcal{D}'$.

Then, we formulate AL procedure as an episodic MDP. Each AL run starts with a small labelled set $\mathcal{L}_0 \subset \mathcal{D}$ along with a large unlabelled set $\mathcal{U}_0 = \mathcal{D} \setminus \mathcal{L}_0$. The following steps are performed at iteration $t$.

1. Train a classifier $f_t$ using $\mathcal{L}_t$.
2. A *state* $s_t$ is characterised by $f_t$, $\mathcal{L}_t$, and $\mathcal{U}_t$.
3. The AL *agent* selects an *action* $a_t \in \mathcal{A}_k$ by following a policy $\pi : s_t \mapsto a_t$ that defines a datapoint $\boldsymbol{x}^{(t)} \in \mathcal{U}_t$ to be annotated.
4. Look up the label $y^{(t)}$ of $\boldsymbol{x}^{(t)}$ in $\mathcal{D}$ and set $\mathcal{L}_{t+1} = \mathcal{L}_t \cup \{(\boldsymbol{x}^{(t)}, y^{(t)})\}$, $\mathcal{U}_{t+1} = \mathcal{U}_t \setminus \{\boldsymbol{x}^{(t)}\}$.
5. Give the agent the *reward* $r_{t+1}$ linked to empirical performance value $\ell_t$.

These steps repeat until a *terminal* state $s_T$ is reached. In the case of *target quality* objective of Sec. 3.1, we reach the terminal state $s_T$ when $\ell_T \geq q$, where $q$ is fixed by the user, or when $T = |\mathcal{U}_0|$. The agent only observes $s_t$, $r_{t+1}$ and a set of possible actions $\mathcal{A}_t$, while $f_t$, $\mathcal{D}'$ and $q$ are the parts of the environment. The agent aims to maximize the *return* of the AL run: $R_0 = r_1 + \ldots + r_{T-1}$ by policy $\pi$ that intelligently chooses the actions, that is, the datapoints to annotate. We now turn to specifying our choice for *states*, *actions*, and *rewards* that reflect the AL objective of minimizing the number of annotations while providing flexibility and transferability.

**States** It only makes sense to perform AL when there is a lot of unlabelled data. Without loss of generality, we can therefore set aside at the start of each AL run a subset $\mathcal{V} \subset \mathcal{U}_0$ and replace $\mathcal{U}_0$ by $\mathcal{U}_0 \setminus \mathcal{V}$. We use the classifier's score $\hat{y}_t$ on $\mathcal{V}$ as a means to keep track of the state of the learning procedure. Then, we take the state representation to be a vector $s_t$ of sorted values $\hat{y}_t(\boldsymbol{x}_i)$ for all $\boldsymbol{x}_i$ in $\mathcal{V}$.

Intuitively, the state representation is rich in information on, for example, the average prediction score or the uncertainty of a classifier. In Fig. 1, we plot the evolution of this vector for $t$ using a policy defined by random sampling, uncertainty sampling, or our learnt strategy, all starting from the same initial state $s_0$. Note that the statistics of the vectors are clearly different. Although their structure is difficult to interpret for a human, it is something RL can exploit to learn a policy.

**Actions** We design our MDP so that taking an action $a_t$ amounts to selecting a datapoint $\boldsymbol{x}^{(t)}$ to be annotated. We characterize a potential action of choosing a datapoint $\boldsymbol{x}_i$ by a vector $\boldsymbol{a}_i$ which consists of the score $\hat{y}_t(\boldsymbol{x}_i)$ of the current classifier $f_t$ on $\boldsymbol{x}_i$ and the average distances from $\boldsymbol{x}_i$ to $\mathcal{L}_t$ and $\mathcal{U}_t$, that is $g(\boldsymbol{x}_i, \mathcal{L}_t) = \sum_{x_j \in \mathcal{L}_t} d(\boldsymbol{x}_i, x_j)/|\mathcal{L}_t|$ and $g(\boldsymbol{x}_i, \mathcal{U}_t) = \sum_{x_j \in \mathcal{U}_t} d(\boldsymbol{x}_i, x_j)]/|\mathcal{U}_t|$, where

$d$ is a distance measure. So, at iteration $t$ we choose an action $a_t$ from a set $\mathcal{A}_t = \{\boldsymbol{a}_i\}$, where $\boldsymbol{a}_i = [\hat{y}_t(\boldsymbol{x}_i), g(\boldsymbol{x}_i, \mathcal{L}_t), g(\boldsymbol{x}_i, \mathcal{U}_t)]$ and $\boldsymbol{x}_i \in \mathcal{U}_t$. Notice, that $\boldsymbol{a}_i$ is represented by the quantities that are not specific neither for the datasets nor for the classifiers.

**Rewards** To model our *target quality* objective of reaching the quality $q$ in as few MDP iterations as possible, we choose our reward function to be $r_t = -1$. This makes the return $R_0$ of an AL run that terminates after $T$ iterations to be $r_1 + \ldots + r_{T-1} = -T + 1$. The fewer iterations, the larger the reward, thus the optimal policy of MDP matches the best AL strategy according to our objective. This reward structure is not greedy because it does *not* restrict the choices of the agent as long as the terminal condition is met after a small number of iterations.

### 3.3 Policy learning using RL

Thanks to our reward structure, learning an AL strategy accounts to finding an optimal (with the highest return) policy $\pi^\star$ of MDP that maps a state $s_t$ into an action $a_t$ to take, i.e. $\pi^\star : s_t \mapsto a_t$. To find this optimal policy $\pi^\star$ we use DQN (Mnih et al., 2013) method on the data that is already annotated. In our case, $Q^\pi(s_t, a_i)$ aims to predict $-(T - t)$: a negative amount of iterations that are remaining before a target quality is reached from state $s_t$ after taking action $a_i$ and following the policy $\pi$ afterwards. Note that it is challenging to learn from our reward function because the positive feedback is only received at the end of the run, thus the credit assignment is difficult.

**Procedure** To account for the diversity of AL experiences we use a collection of $Z$ annotated datasets $\{\mathcal{Z}_i\}_{1 \leq i \leq Z}$ to simulate AL *episodes*. We start from a random policy $\pi$. Then, learning is performed by repeating the following steps:

1. Pick a labelled dataset $\mathcal{Z} \in \{\mathcal{Z}_i\}$ and split it into subsets $\mathcal{D}$ and $\mathcal{D}'$.
2. Use $\pi$ to simulate AL episodes on $\mathcal{Z}$ by initially hiding the labels in $\mathcal{D}$ and following an MDP as described in Sec. 3.2. Keep the *experience* in the form of transitions $(\boldsymbol{s}_t, \boldsymbol{a}_t, r_{t+1}, \boldsymbol{s}_{t+1})$.
3. Update policy $\pi$ according to the *experience* with the DQN update rule.

Even though the features are specific for every $\mathcal{Z}$, the experience in the form of transitions $(\boldsymbol{s}_t, \boldsymbol{a}_t, r_{t+1}, \boldsymbol{s}_{t+1})$ is of the same nature for all datasets, thus a single strategy is learned for the whole collection. When the training is completed, we obtain an optimal policy $\pi^\star$.

In the standard DQN implementation, the Q-function takes a state representation $\boldsymbol{s}_t$ as input and outputs several values corresponding to discrete actions (Mnih et al., 2013), as shown in Fig. 2(a). However, we represent actions by vectors $\boldsymbol{a}_i$ and each of them can be chosen only once per episode as it does not make sense to annotate the same point twice. To account for this, we treat actions as inputs to the Q-function along with states and adapt the standard DQN architecture accordingly, as shown in Fig. 2(b). Then, Q-values for the required actions are computed on demand for $\boldsymbol{a}_i \in \mathcal{A}_t$ through a feed-forward pass through the network. As our modified architecture is still suitable for Q-learning (Watkins & Dayan, 1992; Sutton & Barto, 1998), and the same optimization procedure as in a standard DQN can be used. Finding $\max_{\boldsymbol{a}_i} Q^\pi(\boldsymbol{s}_t, \boldsymbol{a}_i)$ still is possible because our set of actions is finite and the procedure has the same computational complexity as an AL iteration.

**DQN implementation details** RL with non-linear Q-function approximation is not guaranteed to converge, but in practice it still finds a good policy with a few tricks. We use separate target network and experience replay of Mnih et al. (2013), warm start, and prioritized replay of Schaul et al. (2016). Besides, instead of reward normalisation we initialise the bias of the last layer to the average reward that an agent receives in warm start episodes. To compute $Q^\pi(\boldsymbol{s}_t, \boldsymbol{a}_i)$ we use NN where first $\boldsymbol{s}_t$ goes in and a compact representation of it is learnt, then, $\boldsymbol{a}_t$ is added to it and $Q^\pi(\boldsymbol{s}_t, \boldsymbol{a}_i)$ is the output. We use fully connected layers with sigmoid activations except for the final layer that is linear. We perform 1000 RL iterations, each of which consists of 10 AL episodes and 60 updates of the Q-function. As $\hat{y}(\boldsymbol{x}_i)$ we use $p(y_i = 0 | \mathcal{L}_t, \boldsymbol{x}_i)$. The size of $\mathcal{V}$ is set to 30.

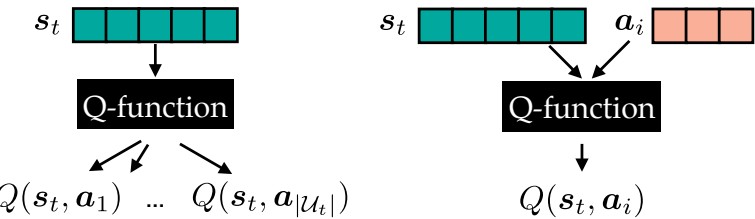

**Figure 2:** Adapting the DQN architecture. Left: In standard DQN, the Q-function takes the state vector as input and yields an output for each discrete action. Right: In our version, actions are represented by vectors. The Q-function takes action and the state as input and returns a single value.

# 4 EXPERIMENTAL EVALUATION

In this section, we demonstrate the transferability and flexibility of our method, as defined in Sec. 3.1, and analyse its behaviour. The corresponding code is publicly available[1].

## 4.1 BASELINES AND PARAMETERS

**Baselines** We will refer to our method as **OURS** and compare it against the following 7 baselines. The first 3 are manually-designed. The next 3 are meta-AL algorithms with open source implementations. The final approach is similar in spirit to ours but no code is available on-line.

**Rs**, random sampling. The datapoint to be annotated is picked at random.

**Us**, uncertainty sampling (Lewis & Gale, 1994), selects a datapoint that maximizes the Shannon entropy $H$ over the probability of predictions: $\boldsymbol{x}^{(t)} = \arg\max_{\boldsymbol{x}_i \in \mathcal{U}_t} H[p(y_i = y \mid \mathcal{L}_t, \boldsymbol{x}_i)]$.

**QUIRE** (Huang et al., 2010), a query selection strategy that uses the topology of the feature space. This strategy accounts for both the informativeness and representativeness of datapoints. The vector that characterizes our actions is in the spirit of this representativeness measure.

**ALBE** (Hsu & Lin, 2015), a recent meta-AL algorithm that adaptively combines strategies, including **Us**, **Rs** and **QUIRE**.

**LAL-ind** (Konyushkova et al., 2017), a recent approach that formulates AL as a regression task and learns a greedy strategy that is transferable between datasets.

**LAL-iter** (Konyushkova et al., 2017), a variation of **LAL-ind** that tries to better account for the bias caused by AL selection.

**MLP-GAL(Te)** (Pang et al., 2018), a recent method that learns a strategy from multiple datasets with a policy gradient RL method.

**Parameters** We use logistic regression (LogReg) or SVM as our base classifiers for AL. We make no effort to tune them and use their sklearn python implementations with default parameters. This corresponds to a realistic scenario where there is no obvious way to choose parameters.

Recall from Sec. 3.2, that our strategy is trained to reach the target quality $q$. For each dataset, we take $q$ to be 98% of the maximum quality of the classifier trained on 100 randomly drawn datapoints, which is the maximum number of annotations we allow. We allow for a slight decrease in performance (98% instead of 100%) because AL learning curves usually flatten and our choice enables AL agents to reach the desired quality much quicker during the episode. We use the *same* RL parameters in all the experiments and also describe additional details in the appendix.

## 4.2 TRANSFERABILITY

We tested the transferability of **OURS** on 10 widely-used standard benchmark datasets from the UCI repository (Dheeru & Karra Taniskidou, 2017): 0-*adult*, 1-*australian*, 2-*breast cancer*, 3-*diabetes*, 4-*flare solar*, 5-*heart*, 6-*german*, 7-*mushrooms*, 8-*waveform*, 9-*wdbc*. We use LogReg and ran 500 trials where AL episodes run up to 100 iterations.

---

[1]`https://github.com/author/project`, to be updated when the paper is public.

| Scenario | test | leave-one-out | | | | | | | | |
|---|---|---|---|---|---|---|---|---|---|---|
| **Dataset** | 0 | 1 | 2 | 3 | 4 | 5 | 6 | 7 | 8 | 9 |
| **Rs** | *50.78* | 25.31 | 25.65 | 30.33 | 15.57 | 44.83 | 20.80 | 42.81 | 45.28 | 19.36 |
| **Us** | 41.83 | **13.53** | 27.07 | *27.84* | 15.50 | 37.10 | **15.60** | **15.60** | 23.83 | **7.25** |
| **QUIRE** | 58.33 | 30.02 | 33.33 | 37.12 | **9.02** | 57.58 | 20.30 | 42.9 | *36.49* | 15.45 |
| **ALBE** | 55.66 | 29.79 | 31.84 | 33.62 | 10.91 | 50.71 | 21.02 | 39.12 | 41.23 | 16.16 |
| **LAL-ind** | 59.39 | *20.88* | 20.85 | **26.63** | 15.31 | 44.14 | *18.16* | *24.15* | 39.13 | *11.22* |
| **LAL-iter** | 63.29 | *20.24* | 21.79 | 28.03 | 14.84 | *40.38* | 19.90 | 25.2 | 36.97 | *10.39* |
| **OURS** | **37.52** | **14.15** | **18.79** | **26.77** | *14.67* | **32.16** | **15.06** | 21.94 | **20.91** | **7.09** |
| **notransf** | — | 15.01 | 16.14 | 24.40 | — | 23.26 | 14.65 | 16.47 | 18.06 | 7.14 |

**Table 1:** Average number of annotations required to reach a predefined quality level.

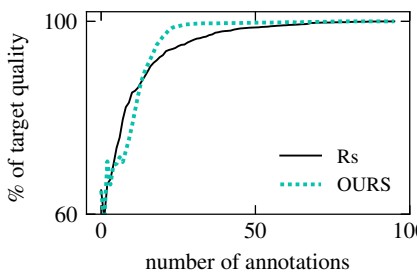

**Figure 3:** Example of non-myopic behaviour of a learnt RL strategy

| Baseline | Rs | Us | OURS |
|---|---|---|---|
| LogReg-100 | 32.07 | −28.80% | **−34.71%** |
| LogReg-200 | 80.06 | −29.61% | **−39.96%** |
| LogReg-500 | 51.59 | −31.49% | **−37.75%** |
| SVM | 30.87 | −7.81% | **−28.35%** |

**Table 2:** Increasing the number of annotations still using logistic regression (first three rows) and using SVM instead of logistic regression as the base classifier (fourth row). We report the average number of annotations required using **Rs** and the percentage saved by either **Us** or **OURS**.

In Table 1 we report the average number of annotations required to achieve the desired target accuracy using either our method or the baselines. In the 9 columns marked as *leave-one-out*, we test out method using a leave-one-out procedure, that is, training on 8 of the datasets selected from number 1 to number 9, and evaluating on the remaining one. In the course of this procedure, we never use dataset 0-*adult* for training purposes. Instead, we show in the column labelled as *test* the average number of annotations needed by all 9 strategies learnt in the leave-one-out procedure (the standard deviation is 2.34). In each column, the best number appears in **bold**, the second is underlined, and the third is printed in *italics*. We consider a difference of less than 1 to be insignificant and the corresponding methods to be *ex-aequo*.

**OURS** comes out on top in 8 cases out of 10, second and third in the two remaining cases. As it has been noticed in the literature, **Us** is good in a wide range of problems (Konyushkova et al., 2017; Pang et al., 2018). In our experiments as well, it comes second overall and, for the same level of performance, it saves 29.80% over **Rs** while **OURS**, saves 34.71%. Table 2 reports similar results reaching 98% of the quality of a classifier trained with 200 and 500 random datapoints instead.

Unfortunately, we cannot compare **OURS** to **MLP-GAL(Te)** in the same fashion for lack of publicly available code. They report results for 20 annotations, we therefore check that even if we also stop all our episodes that early, **OURS** still outperforms the strongest baseline **Us** in 90% of cases whereas **MLP-GAL(Te)** does so in 71% of the cases. Besides, we learn a policy using 5 times less data: 10 000 AL episodes instead of 50 000.

### 4.3 FLEXIBILITY

To demonstrate the flexibility of our approach now we repeat the experiments of Sec. 4.2 with our method, best baseline and random sampling using an SVM instead of LogReg and report the results in the last row of Table 2. Note that **Us** saves only 8% with respect to **Rs**, which is much less than in the experiments of Sec. 4.2 shown in rows 1 to 3. This stems from the fact that the sklearn implementation of SVMs relies on Platt scaling (Platt, 1999) to estimate probabilities, which biases the probability estimates when using limited amounts of training data. By contrast, **OURS** is much less affected by this problem and delivers a 28% cost saving when being transferred across datasets.

As predicted probabilities of SVM are unreliable during early AL iterations, greedy performance maximization is unlikely to result in good performance. It make this setting a perfect testbed to

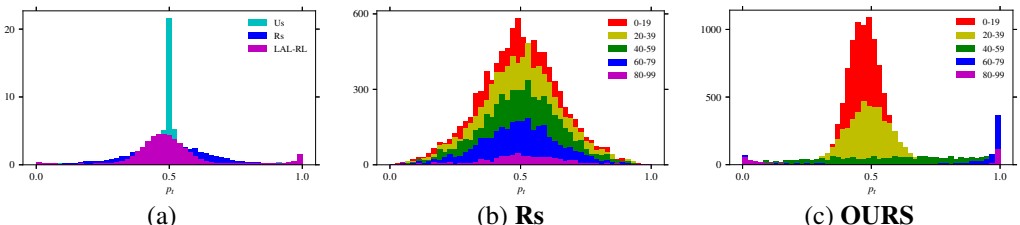

**Figure 4:** Comparing the behavior of **Rs**, **Us** and **OURS**. (a) Histogram of $p_t$ for **Rs** in blue, **Us** in cyan, and **OURS** in purple. (b) Evolution over time for random. (c) Evolution over time for **OURS**.

validate the non-myopic strategies *can* be learned by **OURS**. In Fig. 3 we plot the percentage of the target quality reached by **Rs** and **OURS** as a function of the number of annotated datapoints on one of the UCI datasets. The curve for **OURS** demonstrates a non-myopic behaviour. It is worse than **Rs** at the beginning for approximately 15 iterations but almost reaches the target quality after 25 iterations, while it takes **Rs** 75 iteration to catch up.

### 4.4 ANALYSIS

We now turn to analysing the behaviour of **OURS** and its evolution over time. To this end, we ran additional experiments to answer the following questions.

**What do we select?** While performing the experiments of Sec. 4.2 we record $p_t = p(y^{(t)} = 0|\mathcal{L}_t, x^{(t)})$. We show the resulting normalized histograms in Fig. 4(a) for **Rs**, **Us**, and **OURS**. The one for **Rs** is very broad and it simply represents the distribution of available $p_t$ in our data, while the one for **Us** is very peaky as it selects $p_t$ closest to 0.5 by construction. Figs. 4 (b,c) depicts the evolution of $p_t$ for **Rs** and **OURS** for the time intervals $0 \le t \le 19, 20 \le t \le 39, 40 \le t \le 59, 60 \le t \le 79, 80 \le t \le 99$. The area of all histograms decreases over time as episodes terminate after reaching the target quality. However, while their shape remains roughly Gaussian in the **Rs** case, the shape changes significantly over time in case of **OURS** strategy. Evidently, **OURS** starts by annotating highly uncertain datapoints, then switches to uniform sampling, and finally exhibits a preference for $p_t$ values close to 0 or 1. In other words, the **OURS** demonstrates a structured behaviour.

**Transfer or not?** To separate the benefits of learning a strategy and the difficulties of transferring it, we introduce an artificial scenario **OURS-notransfer** in which we learn on one-half of a dataset and transfer to the other half. In Table 1 **OURS-notransfer** is better than **OURS** in 3 case, much better in 2 and equal in 3 (we skip one small dataset). This shows that having access to the underlying data distribution confers a modest advantage to **OURS**. Therefore, our approach still enables to learn a strategy that is competitive to having access to the underlying distribution thanks to its experience on other AL tasks. We also check how **OURS-notransfer** performs on unrelated datasets, for example, learning the strategy on dataset 1 and testing it on datasets 2-9. The success rate in this case drops to around 40% on average, which again confirms the importance of using multiple datasets. As learning on one dataset to apply to another does not work well in general, we conclude that **OURS** learns to distinguish between datasets to be successful across datasets.

## 5 CONCLUSION

We have presented a data-driven approach to AL that is transferable and flexible. It can learn strategies from a collection of datasets and then successfully use them on completely unrelated data. It can also be used in conjunction with different base classifiers without having to take their specificities into account. The resulting AL strategies outperform state-of-the-art approaches. Our AL formulation is oblivious to the quality metric. In this paper, we have focused on the accuracy for binary classification tasks, but nothing in our formulation is specific to it. It should therefore be equally applicable to multi-class classification and regression problems. In future work, we plan to generalize it to these additional ML models. Another interesting direction is to combine learning *before* the annotation using meta-AL on unrelated data and *during* the annotation to adapt to specificities of the dataset.

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

# Appendices

## A  EXPERIMENTAL SETUP

We provide additional details about our experimental setup and the parameter values necessary to replicate our results.

**AL parameters**  The parameters of LogReg and SVM classifiers were left to their default values in the sklearn package. For LogReg, they include $l^2$ penalty with regularization strength 1 and a maximum of 100 iterations. For SVM the most important parameters include rbf kernel and penalty parameter of 1. The distance measure $d$ between datapoints is the cosine distance.

**RL parameters**  The RL procedure starts with 100 "warm start" episodes with random actions and 100 Q-function updates. While learning an RL policy, the Adam optimizer is used with learning rate 0.0001 and a batch size 32. To force exploration during the course of learning, we use $\epsilon$-greedy policy $\pi$, which means that with probability $1 - \epsilon$ the action $a_t = \arg\max_a Q_\theta^\pi(s_t, a)$ is performed and with probability $\epsilon$ a random one is. The parameter $\epsilon$ decays from 1 to 0 in 1000 training iterations. We incorporate the following techniques: 1) separate target network (Mnih et al., 2013) to deal with non-stationary targets (update rate 0.01), 2) replay buffer (Mnih et al., 2013) (of size 10 000) to avoid correlated updates of neural network, 3) prioritized replay Schaul et al. (2016) to use the experience from the replay buffer with the highest temporal-difference errors more often (the exponent parameter is 3).

**LAL baselines**  The baselines **LAL-iter** and **LAL-ind** are not *flexible* as they were originally designed to deal with Random Forest classifiers. In order we use them within our experimental setup with LogReg, we let them train 2 classifiers in parallel and use the hand-crafted by Konyushkova et al. (2017) features of RF in AL policy.

## B  ADDITIONAL RESULTS

In Sec 4.2, we reported the average duration and variance of the episodes of 9 learned strategies **OURS** on dataset 0-*adult*. The individual durations for all strategies are 38.80, 37.72, 36.74, 33.95, 34.58, 38.76, 37.46, 41.84, and 37.85. Fig. 5 shows the learning curves for all the baselines and for the 9 strategies. Some variability is present, but in 8 out of 9 cases **OURS** outperforms all others baseline and once it shares the first rank with **Us** in terms of average episode duration.

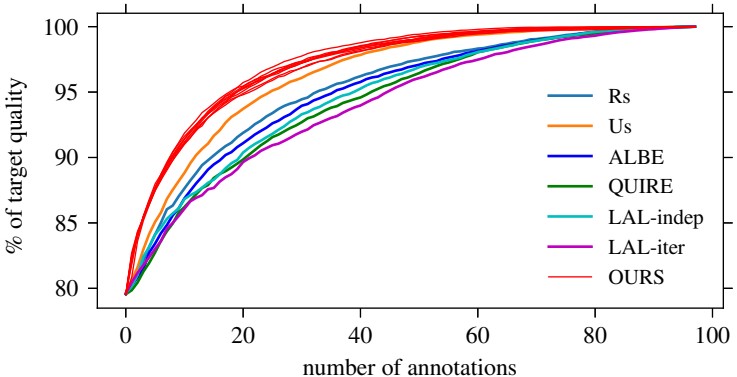

**Figure 5:** Performance of all the strategies on 0-*adult* dataset.

**Learning curves**  Figs. 6 and 7 show additional learning curves that depict the performance of our baselines with LogReg and SVM. The LogReg experiment shows the curves for all the methods and SVM show the curves for the 2 methods that delivered the best performance on average in the experiments of Sec. 4.2 and random sampling. Note that the SVM curve with dataset 5-*flare solar* also clearly exhibits non-myopic behavior.

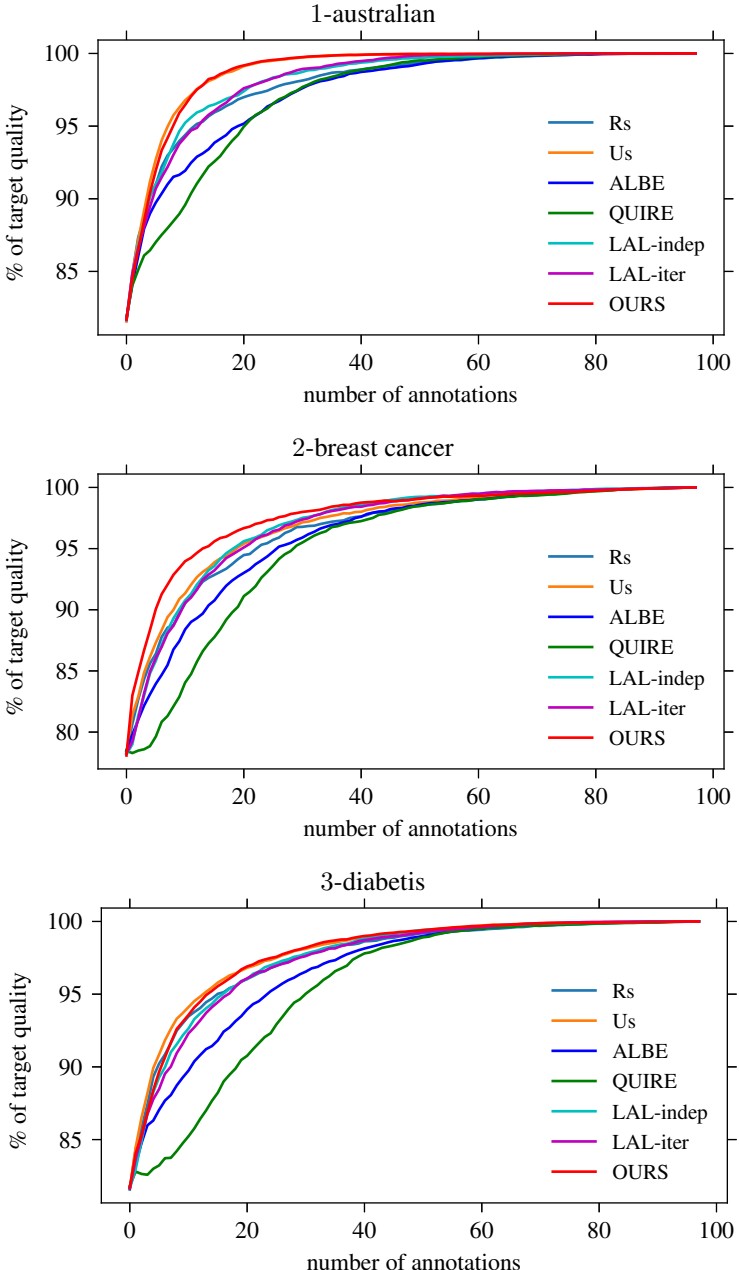

**Figure 6:** Results of experiment from Sec. 4.2. Performance of all baseline strategies on 3 first datasets.

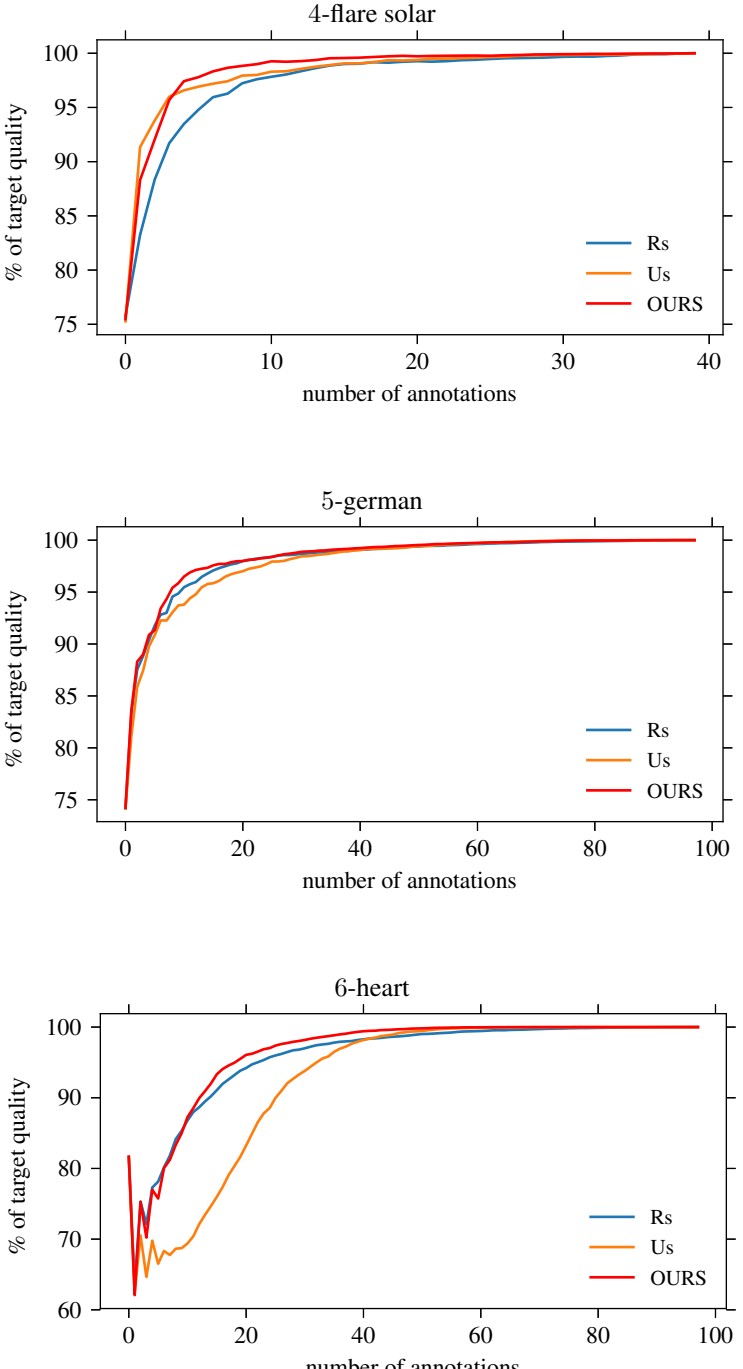

**Figure 7:** Results of experiment from Sec. 4.3. Performance of 3 top strategies from experiment of Sec. 4.2 and a random sampling on the next 3 datasets.

