# OpenReview forum: "Discovering General-Purpose Active Learning Strategies"
_ICLR.cc/2019/Conference_

### Official Review · AnonReviewer1 · 2018-11-02
**This paper describes the use of reinforcement learning to learn active learning strategies. This paper attempts to increase the scope of the learning of active learning strategies to transfer across very different datasets.**

**Rating:** 4
**Confidence:** 4

**Review:**

This paper presents a laudable attempt to generalize the learning of active learning strategies to learn general strategies that apply across many different datasets that have variables of different, not pre-determined, types, and apply the learned active learning strategies to datasets that are different from what they have been learned with. The paper is written quite clearly and is clear in its discussion of what its advance is beyond the current state of the art.

Unfortunately, the motivation of the details of the algorithm and the experiment analysis leave the paper short of what is needed to truly assess the value of this area of work and; therefore, short of what is needed for publication in ICLR. The most notable shortcoming is on page 4, at the bottom, where the actions are described. Among the components of the actions are statistics related to the dataset---the average distance from the chosen point to all the labeled data, and the average distance from the chosen point to all the unlabeled data. The authors do not provide a motivation for the use of these particular statistics. Additionally, the authors did not explore any other statistics. I should think that statistics relevant to the sparsity of the data (e.g., how well they cluster). Additionally, what distance measure is being used? A variety of distance metrics should be explored, such as d-separation for continuous variables and Hamming distance for discrete variables, should be tested, as they intuitively seem likely to affect the results. Additionally, many values are chosen for the experiments without motivation and without testing a variety of values (e.g., 30 for the size of the dataset used to calculate the reward, 1000 RL iterations, and others).

In the experiments, there needs to be discussion of how much variety there is in the different datasets in terms of their statistical properties that are relevant to active learning, such as how well the data cluster? That would help in understanding why the new algorithm performs as it does relative to the baseline.

One relatively minor point: The authors state on page 3, "For example, the probability that the classifier assigns to a datapoint suits this purpose because most classifiers estimate this value." This is a bit misleading---only generative classifiers would do this, not discriminative classifiers.

Pros:
1. Very clear writing.
2. Good motivation for the general problem.
3. Precise description of algorithm.

Cons:
1. Poor motivation for the particular algorithm implementation---features used in the actions, parameter values chosen.
2. Lack of experiments with different choices for features and parameter values.
3. Lack of assessment of the dataset characteristics and how they relate to algorithm performance.

---

### Official Review · AnonReviewer3 · 2018-11-02
**OK paper. Well written, but weak novelty.**

**Rating:** 4
**Confidence:** 5

**Review:**

Summary: This paper studies the recently problem of learning active learning (LAL). It sets up a MDP where the the state is determined by the labeled, unlabelled datasets and classifier, the acton is to query a point, the reward is linked to classifier test set performance improvement and the transition is to update the base classifier. Recent Q-learning algorithms are used to perform the optimisation. The results show that it outperforms some classic handcrafted AL algorithms and some prior LAL algorithms. A feature of this paper is that the method is relatively simple compared to some prior LAL methods, and also that it learns policies that can transfer successfully across diverse heterogenous datasets.

Strengths:
+ Good results.
+ Nice that it works well while being simpler and faster than prior transferrable method MLP-GAL.
+ Generally well written.
+ Fig 4 is interesting.

Weaknesses:
- Novelty/originality is rather incremental.
- Experiments are still on toy datasets.

Specifics:
1. Novelty: The concept of formulating AL as a MDP for optimisation is now a standard idea. The optimisers used are recent off-the-shelf Q-learners. The result is that this method is similar to a non-myopic extension of LAL (Konyushkova’17) but several papers already did non-myopic AL. In particular it’s very similar to the SingleRL method in (Pang’18). The only differences are smallish design parameters like: slightly different reward function definition, use Q-learning instead of policy-gradient optimiser, and slightly different state featurisation. The improved sample/speed-efficiency vs SingleRL is likely relatively automatic due to use of recent Q-learning optimisers, rather than vanilla PG optimiser of SingleRL. Not clear that benefit comes from something uniquely contributed here. Other limitations of various prior LAL work, such as binary classifier only, are not alleviated here.
2. Experiments: The experiments are on toy datasets. Particularly given the small novelty, then evaluation should be much more. For example: 1. How well does it work when transferred to a relatively less toy dataset such as CIFAR. 2. To what extent can it transfer across classifiers rather than only across datasets?
3. The state representation as a sorted list of scores is rather unintuitive. Is there any intuition on what smart decisions the model could be using this to make?
4. The featurisations used are not very standard: Like the classifier state sorted score list, and the action featurisation (instance score, instance distance to class, instance distance to unlabelled). It would be good to evaluate this featurisation with a supervised active learner (like LAL), in order to disambiguate whether the good performance comes from these feature choices, or from the recent RL algorithms used to optimise. Similarly for the choice of reward function.
5. How does the proposed method deal with a suite of training datasets for AL that are of greatly varying difficulty. A relatively very easy dataset needing << 100 examples to reach threshold would generate few AL training examples due to early stopping. A very hard dataset might use all 100 examples. Does it mean that easy datasets contribute less to training than hard ones?

---

> ### Comment · AnonReviewer3 · 2018-12-08
> **Thanks for the response**
>
> Thanks to the authors for their clarifications.
>
> In general my feeling is still "not quite good enough" on novelty grounds.
>
> Given the similarities in the current approach and various recent papers, the analysis and evaluation should be top notch to quality for a top tier ICLR publication.
>
> Various issues design choices and parameters could be better justified and evaluated as highlighted by other reviewers.
> Also, when I mentioned CIFAR, I did not mean transfer across categories within CIFAR. I meant include datasets like CIFAR among the training datasets. And/Or show that once trained on the UCI datasets used, evaluate whether it can transfer to CIFAR. This might work given the type of representations used in this paper, and would be a clearer plus on Pang'18 whose dataset embedding is not ideal for images.
> Also, when I said transfer across classifiers. I meant evaluate whether a model trained for SVM can successfully test on LogReg, and vice-versa, etc.  Not whether a unique RL model can be trained for each, as seems to be the case in the current result.

---

### Official Review · AnonReviewer2 · 2018-11-09
**A Reinforcement Learning approach to Active Learning**

**Rating:** 4
**Confidence:** 5

**Review:**

The authors suggest to model active learning (AL) as a Markov Decision Process to try to learn the best possible AL strategy across related domains.

The paper is well-written and structured -- although the background section could be expanded. Sec 3 presents the method in a clear and straightforward manner.

My main concern with regards to the paper is novelty. The authors mention two main contributions, the first one being to defined the AL objective to minimize the number of annotations required to achieve a given prediction quality, instead of maximizing performance given an annotation budget. There has been AL approaches from that perspective in the past (e.g., https://arxiv.org/pdf/1510.02847.pdf).

The second contribution has to do with a procedure to learn the AL strategy using data from different domains (with available labels). Again, the literature in transfer learning in Reinforcement Learning is extensive and should be discussed.

---

### Official Review · AnonReviewer4 · 2018-11-17
**An intuitive combination of reinforcement learning and active learning.**

**Rating:** 5
**Confidence:** 4

**Review:**

Summary:
This paper presents an RL approach to active learning that is generic across ML model being learned, and across dataset being used. The paper formulates the standard active learning problem as an MDP with the objective of minimizing the number of annotated labels required to meet a pre-specified prediction quality.

The MDP state proposed by this paper is the current performance score on each sample in a hold-out set. The actions are specified by selecting a datapoint from the set of all un-annotated datapoints. The action feature vector consists of the current performance score of the model on the datapoint, and the average distance of that datapoint from every datapoint in the labeled set and every datapoint in the unlabeled set.

Review:
I do not recommend this paper for publication in ICLR because I believe:
1) the work is too incremental
2) the comparison to baseline and competing methods is incomplete
3) some design decisions of the proposed method are not well motivated.

I appreciated the clarity of the writting, and the paper organization. I also believe that the proposed method is quite intuitive, and is a good addition to the field. Finally, I appreciate that sufficient experimental details are available within the paper to be able to easily reproduce the results.

Details:
My points (1) and (2) are highly related, so I will discuss both simultaneously. I find that this paper makes only incremental forward progress from the Pang 2018 paper and the Konyushkova 2017 paper. The methodology here looks very similar to the SingleRL method, which Pang 2018 notes can be considered a special case of Konyushkova 2017's method. I think that the work in this paper would be sufficient to stand on its own if it performed a convincing comparison to SingleRL and/or MLP-GAL from Pang 2018. I recognize that this paper references why no such comparison currently exists, but I think this comparison would be extremely valuable to the paper.

A further comment on my point (2), I do not find the comparisons to baseline methods to be entirely convincing. Of note, only the average performance for each method is reported. I'm curious of the variance---and more specifically the standard error and number of independent runs---of each of the reported results. On many of the datasets, the performance difference between the proposed method and uncertainty sampling is quite small in table 1.

A final comment on point (2): I would have liked to see more exploration of different models. I think table 2 is quite informative, showing notable differences between simple baseline AL methods. I would have liked to see table 2 with more classifiers and with more competing AL methods. Because logistic regression is a simple model, the differences between AL methods may be more subtle. Perhaps a more complex model (say a single hidden layer NN) would show more notable differences.

For point (3), I would have liked to see either an exploration of other design decisions or an explanation of given design decisions. For instance, why only use 30 hold-out samples for the state? I imagine the proposed method would be fairly sensitive to this choice.  Another unexplained design decision was using a maximum budget of 100 datapoints. Table 2 shows some extremely interesting interactions with this budget in its comparison between LogReg-100 and LogReg-200, and further explanation would have been useful. Finally, I would have liked to see some motivation for choice of stopping condition. Using the stopping condition of 98% of maximum performance may have some biasing effect of each method, and it would helpful to have some motivation behind this choice.

Questions:
 - Why did uncertainty sampling have such limited benefits on LogReg-200 in table 2? This was a surprising result to me, as uncertainty sampling consistently outperformed most other methods.
 - Why is there a disparity between the results for the SVM in table 2 and the discussion in the first paragraph of section 4.3?
 - How does choice of final performance metric affect all methods? Choosing final performance to be 98% of maximum performance could have a major effect on each method. Because the proposed method is non-myopic, I would expect that it performs well when this value is large but would perform poorly with a smaller percentage of maximum performance.
 - Is the proposed method sensitive to number of samples used to compute the state?
 - What does figure 1 show? Are the same 30 samples used for all three subfigures? Perhaps this would more interpretable if, instead of showing the predicted class, this figure showed the prediction error.

Minor nitpicks (did not influence decision):
 - The datasets are 1-based indexed sometimes and 0-based indexed sometimes, even with disparities within a single paragraph.
 - Figure 1 appears a long time before it is discussed, which made it difficult to understand what was going on.

---

### Author Response · Authors · 2018-11-26
**Response to common concerns**

We would like to thank the reviewers for their feedback that will help to improve our manuscript. In this reply, we will try address the reviewer’s common concerns on the novelty, representation of states and actions and the algorithm that is chosen for reinforcement learning.

NOVELTY

Active learning with reinforcement learning has become trendy recently, and several works have led to similar ideas with different problem formulations and models. We believe that the literature on this subject is not yet complete and the question of general-purpose AL strategies deserves additional treatment. We would like to mention two features of our method: 1) simplicity and 2) view on the problem from the perspective of budget minimisation given quality constraints. 1) The simplicity of state and action representations in our proposed method make our model conceptually easy to understand and implement and it could be of interest to practitioners. Besides, our method is transparent as it optimises for the same metric that is used for testing. 2) Although the view on the problem as minimising the amount of annotations to reach a given performance was used before in theoretical literature on AL, it opens a new perspective on AL formulation in the form of an MDP with a very simple and domain independent reward function. The reward function is conceptually different from prior work on AL with RL.

STATE AND ACTION

Our state representation is just the simplest representation that can be obtained given a classifier and a dataset. First, we just apply a classifier to validation datapoints. Then, we sort the scores. It is needed because the scores serve as an input to a fully connected layer of neural network that does not allow for permutations. Intuitively, the active learner can extract the information on the confidence of the classifier or proportions of various class predictions. Combined with a score of a potential datapoint (part of action) active learner can estimate from which part of the distribution the datapoint comes from.
In Figure 1 the datapoints are the same in three experiments. We did not use the prediction error in our visualisation. Although it would definitely be more informative than the predicted score, such information is not available to the learner at test time (because validation set in unlabelled).

The action representation in the form of three statistics is again just the simplest representation that relates three components of AL problem: classifier, labelled and unlabelled datasets. It has been noticed by reviewers that two of these statistics are related to the sparcity of data and they represent the heuristic approximation for density of the data. We did not explore other statistics because our primary motivation was to keep our method as simple as possible.

REINFORCEMENT LEARNING ALGORITHM

In fact, we cannot use a standard off-the-self Q-learner, such as DQN procedure because it is not designed to deal with continuous actions which we encounter in our problem formulation. We modified the way how optimisation is performed, instead of having discrete actions, we have actions that are represented by vectors and they serve as an input to DQN instead of being related to several outputs. Although this is a small modification, it was not done in other AL-RL methods which considered only policy gradient optimisation for the task. Adapting DQN for the use in this scenario directly allows to exploit the benefits of bootstrapping for AL task.

---

> ### Author Response · Authors · 2018-11-26
> **Response to common concerns (continuation)**
>
> Finally, we would like to answer the common questions and concerns regarding the choice of parameters and the experimental study.
>
> PARAMETERS
>
> We have chosen 30 datapoints for validation as smaller values result in slightly decreased performance and bigger values do not help to improve the result but make the execution slower. However, the method is not sensitive to this choice of parameter and one value was suitable across all datasets.
> 1000 iterations were chosen by tracking the performance of RL policy on a validation set and stopping the training when the rewards stops growing. In the vast majority of cases it happened before 1000 iterations and thus we set a rule to stop the execution at this point.
>
> EXPERIMENTS
>
> The setting of transfer between various classes of complex datasets (such as CIFAR) is well studied in the literature (Ravi & Larochelle, 2018; Liu et al., 2018; Bachman et al., 2017; Fang et al., 2017; Contardo et al., 2017) and that is why we have chosen to concentrate on transfer between datasets. In this case we need many datasets of comparable difficulty level. Although we have tried to demonstrate the benefits of our method on close-to-realistic settings, we acknowledge that the experiments could be run with even more complex data.
>
> Having classification problems of varying difficulties could lead to some problems contributing more to the training procedure. This is why we performed training on the problems of comparable difficulty (UCI datasets). In practice, the target of general-purpose AL is to perform the best in expectation for any new coming datasets. Then, this bias during training could be beneficial because it concentrates on more difficult cases that can bring more advantage at test time. Nevertheless, as training is performed on datasets of varying difficulty, at test time the AL strategy tries to understand (based on state and action features) how difficult the problems is and the selection policy is adjusted accordingly.
>
> Thank you for bringing our attention to the issue with table 2, the strange behaviour is caused by a typo where we swapped 2-d and 4-th lines of Table 2. With this typo corrected, the explanation in Section 4.3 is consistent with the table.
>
> As stated, we ran 500 trials in every experiment. Often the difference between our method and the second best method is quite small. Despite this fact, the average results in table 2 show the benefit of the proposed method. In addition to this, we ignored small differences in methods (and considered both of them winning) when reporting the ranking results.
>
> Our stopping criteria is motivated by the practical scenario when achieving 98% of prediction quality is enough for a final user if it results in significant cost savings. We varied the stopping conditions when we increase the size of the total target set (98% of quality obtained when trained on 100 datapoint, 200 datapoints or 500 datapoints).

---

> > ### Author Response · Authors · 2018-11-26
> > **Response to minor comments**
> >
> > To conclude, we briefly mention the other smaller comments by the reviewers.
> >
> > Distance measure
> > We used the cosine distance as explained in the experimental setup. It was chosen as its scale is independent of the dimensionality of the data. Other distance measures could be used, but even this simple distance measure allows to reach promising results.
> >
> > "For example, the probability that the classifier assigns to a datapoint suits this purpose because most classifiers estimate this value."
> > By "the probability that the classifier assigns to a datapoint" we mean p(y|x) and not p(x) or p(y,x).
> >
> > Zhang and Chaudhuri, 2015
> > Thank you for bringing the paper by Zhang and Chaudhuri, 2015 to our attention, it is an important piece of work to discuss in the related work. This related work analyses the setting with a strong and weak labeller, while in our work we study a general case with one annotator. Besides, the focus of our work is more on the empirical aspects of AL.
> >
> > Transfer between classifiers
> > So far, we have show that our results hold for logistic regression and SVM (two different families of methods), but it could be interesting to extend it to other classifiers too. Besides, our proposed model can seamlessly transfer to various problems such as regression (by modifying the stopping condition) or multi-class classification (by extending the state representation to a matrix instead of vector), that could be interesting to study as well.
> >
> > Relating to supervised learning
> > It would be interesting to study the contribution of our state and action representation if they are applied with a supervised active learner (like LAL), however, the reward structure is only possible when applied with reinforcement learning.

---

### Meta-Review · Area_Chair1 · 2018-12-10
**More focus is needed on what is novel in this work**

**Confidence:** 4
**Recommendation:** Reject

**Metareview:**

This paper provides further insight into using RL for active learning, particularly by formulating AL as an MDP and then using RL methods for that MDP. Though the paper has a few insights, it does not sufficiently place itself amongst the many other similar strategies using an MDP formulation. I recommend better highlighting what is novel in this work (e.g., more focus on the reward function, if that is key). Additionally, avoid general statements like “To this end, we formalize the annotation process as a Markov decision process”, which suggests that this is part of the contribution, but as highlighted by reviewers, has been a standard approach.